# Phosphate-Buffered Saline and Dimethyl Sulfoxide Enhance the Antivenom Action of Ruthenium Chloride against *Crotalus atrox* Venom in Human Plasma—A Preliminary Report

**DOI:** 10.3390/ijms25126426

**Published:** 2024-06-11

**Authors:** Vance G. Nielsen

**Affiliations:** Department of Anesthesiology, The University of Arizona College of Medicine, Tucson, AZ 85724, USA; vnielsen@anesth.arizona.edu

**Keywords:** ruthenium, dimethyl sulfoxide, phosphate buffer, antivenom, snake venom, thrombelastography, plasma

## Abstract

Ruthenium chloride (RuCl_3_) is widely utilized for synthesis and catalysis of numerous compounds in academia and industry and is utilized as a key molecule in a variety of compounds with medical applications. Interestingly, RuCl_3_ has been demonstrated to modulate human plasmatic coagulation and serves as a constituent of a compounded inorganic antivenom that neutralizes the coagulopathic effects of snake venom in vitro and in vivo. Using thrombelastography, this investigation sought to determine if RuCl_3_ inhibition of the fibrinogenolytic effects of *Crotalus atrox* venom could be modulated by vehicle composition in human plasma. Venom was exposed to RuCl_3_ in 0.9% NaCl, phosphate-buffered saline (PBS), or 0.9% NaCl containing 1% dimethyl sulfoxide (DMSO). RuCl_3_ inhibited venom-mediated delay in the onset of thrombus formation, decreased clot growth velocity, and decreased clot strength. PBS and DMSO enhanced the effects of RuCl_3_. It is concluded that while a Ru-based cation is responsible for significant inhibition of venom activity, a combination of Ru-based ions containing phosphate and DMSO enhances RuCl_3_-mediated venom inhibition. Additional investigation is indicated to determine what specific Ru-containing molecules cause venom inhibition and what other combinations of inorganic/organic compounds may enhance the antivenom effects of RuCl_3_.

## 1. Introduction

Ruthenium chloride (RuCl_3_) is an inexpensive, commonly utilized molecule in industrial, academic, and preclinical medical settings [1,2,3,4,5,6,7,8,9,10,11,12,13,14,15,16,17,18,19,20,21,22,23,24,25,26,27,28,29,30,31,32,33]. RuCl_3_ is a widely used molecule in the synthesis and catalysis of several inorganic and organic compounds under a variety of conditions [1,2,3,4,5,6,7,8,9,10,11,12,13,14,15,16]. RuCl_3_ has also been employed as an agent to enhance imaging in the setting of electron microscopy [17] and the identification of parent ions when using electrospray mass spectrometry [18]. Further, antibiotic [19] and antiviral [20,21,22] properties of RuCl_3_ or its derivatives have been identified. Additionally, antineoplastic effects have been observed in complexes containing or formed from RuCl_3_ [22,23,24]. Interestingly, analgesic agents have recently been synthesized from RuCl_3_ [25,26,27,28,29]. Unexpectedly, RuCl_3_ was found to be a procoagulant agent, enhancing the activation of prothrombin in human plasma exposed to the compound following RuCl_3_ being dissolved in calcium-free phosphate-buffered saline (PBS), but not after RuCl_3_ was dissolved in distilled water [30]. Of particular interest, RuCl_3_ has been identified as a compound that can inactivate a variety of procoagulant and neurotoxic snake venoms in vitro as a sole agent or in combination with other ruthenium-containing compounds [31,32]. Critically, RuCl_3_, as part of a compounded Ru-based antivenom, has been demonstrated to significantly or completely inhibit four different venoms obtained from vipers around the world that are procoagulant or anticoagulant in nature [33,34]. While the precise molecular mechanism by which RuCl_3_ inhibits venom enzymes has not been fully elucidated, it has been posited that ionic compounds formed from RuCl_3_ in PBS interact with activity-critical histidine residues contained within the enzymes, as seen in the inhibition of *Atheris*, *Echis*, and *Pseudonaja* venoms by the Ru-containing compound tricarbonyldichlororuthenium (II) dimer (CORM-2) [35]. Another matter to consider is that RuCl_3_ has been combined with dimethyl sulfoxide (DMSO) at room temperature in solution as a first step in the synthesis of Ru-based chemotherapeutic agents [36]. DMSO is a key component of the antivenom [33,34] used to solubilize carbon monoxide-releasing molecule (CORM-2), which is then dissolved in PBS; therefore, RuCl_3_ and DMSO possibly form another compound within this antivenom. Thus, the specific inhibitory effects on snake venom enzymes of RuCl_3_ in a cationic form by itself, in another ionic compound formed by RuCl_3_ in PBS, or another compound formed from RuCl_3_ and DMSO remained unidentified.

Previous investigations of these compounds were not able to discern the nature of RuCl_3_ ion-mediated inhibition of venom enzymes secondary to two issues. An issue with previous work performed in this laboratory was that snake venom was routinely dissolved in PBS to protect enzymatic activity before and after ultracold storage from potentially adverse acidic pH conditions in water or 0.9% NaCl in water (normal saline, NS). Therefore, there was not a circumstance under which the effects of RuCl_3_ on venom enzymes would occur in the absence of the constituents of the calcium-free PBS utilized in this laboratory with the following characteristics and constituents: pH = 7.4 ± 0.2, 137.9 mM NaCl, 1.47 mM KH_2_PO_4_, 2.67 mM KCl, 8.09 mM Na_2_HPO_4_. In contrast, NS has a pH value of approximately 5.5 with 154 mM NaCl. Another issue suggesting that RuCl_3_ by itself in a cationic form may not play a role in the inhibition of snake venom enzyme activity was that RuCl_3_, in water, had no effect on human plasmatic coagulation whereas RuCl_3_ dissolved in PBS did [30]. Taken together, addressing these issues with a new experimental design was indicated to better define the nature of RuCl_3_-mediated venom enzyme inhibition.

A first step towards further elucidating what possible form of RuCl_3_ ion is responsible for venom enzyme inhibition is the experimental design displayed in Figure 1. Using a well-characterized fibrinogenolytic venom [37,38] obtained from the western diamondback rattlesnake (*Crotalus atrox*) dissolved in either NS or PBS, RuCl_3_ dissolved in distilled water is introduced into the two venom containing solutions. Subsequently, venom, in either diluent, without or with RuCl_3_ addition, is placed into human plasma, with coagulation kinetics assessed with thrombelastography as previously presented [30,31,35]. Similarly, addition of DMSO to NS prior to RuCl_3_ addition serves as the milieu to assess the effects of a RuCl_3_-DMSO-based compound on venom activity. Changes in *C. atrox*-mediated fibrinogenolytic effects on human plasma coagulation kinetics are used to infer the inhibitory effects of Ru-containing ions. The specific thrombelastographic methods used involve the first-derivative parameters displayed in Figure 2. This figure depicts the typical data output derived from a tissue factor-activated human plasma sample, with the time from onset to the maximum rate of clot formation, the maximum velocity of thrombus growth, and the final total clot strength defined by the area under the curve that represents viscoelastic resistance. Thus, utilizing this experimental paradigm and viscoelastic methodology, the first goal of this investigation was to determine if RuCl_3_ as a cationic agent in NS has any inhibitory effect on venom activity in human plasma. The second goal was to determine if RuCl_3_ as a cationic and/or anionic agent in PBS inhibits venom activity. The third goal was to determine if RuCl_3_ exposed to DMSO in NS modulates venom activity differently than RuCl_3_ alone, which would be indicative of a novel DMSO-Ru-containing antivenom agent. Lastly, the fourth goal was to document the basic appearance of the various fluids and combinations of RuCl_3_ and DMSO additions at room temperature over time and discern differences in color and precipitate formation.

## 2. Results

### 2.1. Effects of NS and PBS on RuCl_3_ Antivenom Efficacy

Data generated from this series of experiments are displayed in Figure 3. All venom-containing conditions involved a five-minute incubation at room temperature of venom and NS, PBS, or either fluid containing RuCl_3_. Compared to tissue factor-activated human plasma not exposed to any additions (control), samples containing venom dissolved in NS or PBS significantly increased the time to maximum velocity of clot growth (TMRTG), decreased the velocity of clot growth (MRTG), and decreased clot strength (TTG) values without any difference in venom effects secondary to the diluent. Plasma samples containing venom in PBS with RuCl_3_ addition had TMRTG and TTG values not different from control plasma, but MRTG values were significantly decreased, albeit to a very small extent compared to control plasma. Venom exposed to PBS and RuCl_3_ had effects on all three coagulation kinetic parameters significantly inhibited compared to venom not exposed to RuCl_3_ in PBS. Regarding venom in NS exposed to RuCl_3_, TMRTG values were significantly greater than those seen in control plasma but significantly smaller than in plasma exposed to venom in NS without RuCl_3_ addition. Further, venom in NS exposed to RuCl_3_ had MRTG values significantly smaller than those seen in control plasma but significantly greater than in plasma exposed to venom in NS without RuCl_3_ addition. Of interest, TTG values were not different from control plasma in venom in NS exposed to RuCl_3_ but significantly greater than venom in NS without RuCl_3_ addition. Venom in NS exposed to RuCl_3_ inflicted significantly greater increases in TMRTG values and decreases in MRTG values in plasma compared to those generated by venom exposed to RuCl_3_ in PBS. Lastly, there was no difference between TTG values in plasma exposed to venom and RuCl_3_ in PBS or NS.

### 2.2. Effects of DMSO in NS on RuCl_3_ Antivenom Efficacy

Data generated from this series of experiments are displayed in Figure 4. As with the previous series of experiments, venom was incubated for five minutes with the indicated additives, all within NS as the diluent. All conditions presented were obtained from a fresh, separate series of experiments that used the same lot of human plasma and the same lot of venom dissolved in NS. Compared to control plasma, venom again significantly prolonged TMRTG values, decreased MRTG values, and diminished TTG values. As before, exposure to RuCl_3_ resulted in a significant inhibition of venom-mediated anticoagulation, but the restoration of coagulation was still only partial compared to control condition values. Critically, while exposure of venom to RuCl_3_ and DMSO resulted in TMRTG values that were not significantly different from one another and still significantly greater than control values, the combination of RuCl_3_ and DMSO caused significantly greater MRTG and TTG values than in venom samples exposed to RuCl_3_ alone. Further, the combination of RuCl_3_ and DMSO resulted in MRTG and TTG values significantly lower than control values. When considered in composite, these data support the notion that a new, more effective compound is formed from RuCl_3_ and DMSO or that, in some way, DMSO enhances the solubility of RuCl_3_ or the affinity of RuCl_3_ binding to venom enzymes. 

### 2.3. Appearance of DMSO and RuCl_3_ in NS and PBS over Time

These experiments were designed to document the physical appearance of various solutions without or with the addition of RuCl_3_ and DMSO. A milliliter of the indicated solutions was placed in clear Eppendorf plastic tubes placed on their sides at 25 °C, as displayed in Figure 5. No precipitate was present at thirty minutes. Except for a pale silver hue in samples containing RuCl_3_, there were no differences between the samples.

## 3. Discussion

A variety of unexpected results were generated by these experiments. First, the data generated in samples involving NS as the experimental vehicle support the concept that RuCl_3_ in a cationic form is a significant inhibitor of the fibrinolytic enzymes present in *C. atrox* venom. These results are in opposition to the lack of effect on human coagulation by a presumed single cationic form of RuCl_3_ [30]. However, of greater interest was the finding that RuCl_3_ in PBS was a superior inhibitor of *C. atrox* venom activity compared to RuCl_3_ in NS, but only to a certain, albeit significant, extent. While TMRTG and MRTG values were best preserved by RuCl_3_ in PBS compared to the compound in NS, there was no significant difference in TTG values. Given the various phosphate-containing compounds in PBS (Figure 1), it is entirely possible that at least one other Ru cation or anion formed from RuCl_3_ is responsible for the additional inhibition of *C. atrox* venom activity in addition to a simple cation formed from the loss of Cl from RuCl_3_. Given the inorganic chemistry involved, not only two, but several different Ru complexes may be formed and at play. RuCl_3_ may certainly form complexes with phosphates, sulfur-containing amino acids, histidine, or combinations of those. Furthermore, the pH should have an effect on the nature of the coordinated water molecules (including, e.g., an equilibrium of the type [Ru-OH_2_]n <-> [Ru-OH]n−1 + H^+^). Another possibility for enhanced inhibition of venom activity by RuCl_3_ in the PBS microenvironment may be that the pH of the solution simply enhances the interaction of a simple Cl alone-containing cation formed by the loss of Cl from RuCl_3_ with critical amino acid residues of the fibrinogenolytic enzymes contained in the venom. Taken as a whole, it is noteworthy that there very well may be several ions derived from RuCl_3_ in PBS that inhibit *C. atrox* activity.

Another remarkable finding was that RuCl_3_ likely formed a compound containing DMSO in NS during the brief five-minute incubation that was significantly more effective than RuCl_3_ alone. Again, one must also consider the possibility that the solubility of RuCl_3_ was enhanced (perhaps by micro-suspensions) or the affinity of RuCl_3_ binding to venom enzymes could have occurred as an alternative explanation for the phenomena observed. While the increase in protection of coagulation from *C. atrox* venom by this DMSO-RuCl_3_ compound was not that large in this biochemical paradigm, it should be noted that the concentration of RuCl_3_ in the final mixture of antivenom utilized successfully in vivo was 500 µM [33,34]—five times that of the in vitro assay. Further, the concentration of DMSO in the antivenom used in vivo was 10%—ten times the concentration used in the present study. The reason to use the 100 µM concentration of RuCl_3_ in in vitro studies is that when diluted one hundred-fold in the final assay, a concentration of 1 µM RuCl_3_ does not affect coagulation [30]. However, greater concentrations of RuCl_3_, which would include 5 µM (following a hundred-fold dilution), enhance human coagulation by increasing prothrombin activation [30]. When considered in total, while it cannot be documented in vitro secondary to interference with the thrombelastographic assays, it is possible that there is at least a five times greater generation of a DMSO-RuCl_3_ compound in the antivenom utilized in vivo effectively against four hemotoxic snake venoms [33,34]. In addition to being in a greater concentration when administered in the antivenom used in vivo, other potential biochemical features of the DMSO-RuCl_3_ compound may enhance venom inhibition via pharmacokinetic effects. Specifically, the DMSO-RuCl_3_ compound may be more lipid-soluble than the RuCl_3_ and RuCl_3_-phosphate ions, allowing greater interactions with venom enzymes that may exist in the interstitial space separated from the immediately adjacent area of antivenom injection. The DMSO-RuCl_3_ compound could pass through basement membranes or between cell membranes in a more facile fashion via superior lipid solubility. In summary, while speculative, it is possible that this newly recognized DMSO-RuCl_3_ compound may play a more substantive role in ruthenium antivenom-mediated treatment of hemotoxic envenomation than these in vitro findings may suggest.

The experiments documenting the appearance of RuCl_3_-containing solutions were designed to determine potential stability issues with the combinations with varying pH, phosphate content, or DMSO presence. The pale silver color is what has been seen when formulating the antivenom for administration to envenomed rabbits over a five-minute period [33,34], with the exception from the present study being a more intense shade given the 500 µM concentration of RuCl_3_. The author also noted that addition of CORM-2 in DMSO to the RuCl_3_-containing solution resulted in a still translucent but somewhat more yellow solution, again without precipitate forming within the five minutes of formulation. In the present investigation, even 30 min at room temperature did not change the physical appearance of any of the combinations of compounds in Figure 5. When these observations are considered together, it seems likely that whatever compounds are being formed from RuCl_3_ remain in solution and may be stable for at least the time of observation.

The precise snake venom enzyme target sites that are vulnerable to interaction with RuCl_3_-derived ions and compounds remain to be identified. However, it is quite possible that the same critical amino acid residues vulnerable to binding to the ruthenium radical formed by CORM-2 after release of carbon monoxide may also be binding sites for RuCl_3_-derived ions and compounds. Specifically, histidines critical to venom function have been identified as sites for CORM-2 radical interactions, inactivating enzymes such as *Apis mellifera* phospholipase A_2_ [39] and several other biochemically diverse hemotoxic snake venoms in vitro [35]. CORM-2-derived ruthenium radicals also bind to thiol groups [40]. A preliminary investigation that sought to bridge the gap in knowledge of vulnerability to CORM-2-derived ruthenium radicals and RuCl_3_-derived compounds involved the documentation of inhibition of a phospholipase A_2_ derived from the venom of the Mojave rattlesnake with RuCl_3_ in PBS [32], the same toxin demonstrated to be inhibited by CORM-2-derived ruthenium radicals in an earlier work [41]. Of interest, it was determined that RuCl_3_ in PBS and CORM-2-derived ruthenium radicals sometimes separately inhibited hemotoxic snake venoms, sometimes only one rather than both compounds inhibited venom activity, and sometimes the two compounds separately inhibited venom activity, but when combined, inhibited venom activity to a far greater extent than either one alone [31]. These investigations are suggestive of several possible manners in which ruthenium-derived compounds may inhibit venom enzymes. For example, the same critical-to-function amino acid may have different binding kinetics with different ruthenium-containing ions or radicals on any specific enzyme. Another possibility is that there may be more than one amino acid that is key to enzymatic activity that has greater affinities to specific ruthenium-containing compounds. In conclusion, the mixture of RuCl_3_-derived ions and DMSO-RuCl_3_ compound may have very similar amino acid or thiol targets critical to enzymatic function in hemotoxic snake venoms.

The present investigation has a number of limitations. First, only one venom was utilized to achieve the goals of the present work, and other venoms with different proteomes might be expected to potentially have a different antivenom response (vulnerability and degree of inhibition) to the various RuCl_3_-derived compounds identified. *C. atrox* venom was utilized as it allows facile degradation of fibrinogen in sodium citrate anticoagulated plasma secondary to its fibrinogenolytic activity being calcium independent [38]. This biochemical characteristic of *C. atrox* venom allowed for a consistent, rapid, and marked compromise of coagulation that was subsequently utilized to assess the antivenom efficacy of the various RuCl_3_-derived compounds. To address this limitation, future investigation is planned with the paradigm described here with other venoms such as those of coral snakes (e.g., *Micrurus fulvius*, *Micrurus tener*) or the Mojave rattlesnake (*Crotalus scutulatus scutulatus*, venom type A) that are the phospholipase A_2_-resplendent [39,41]. Revisiting procoagulant venoms previously demonstrated to be inhibited by RuCl_3_ to some extent (e.g., *Bothrops moojeni*, *Calloselasma rhodostoma*, *Echis leucogaster*, *Heloderma suspectum*, *Oxyuranus microlepidotus*, *Pseudonaja textilis*) that contain metalloproteinases, serine proteases, and kallikrein-like activity [31] could also provide evidence of similar or different degrees of inhibition from the variety of RuCl_3_-derived compounds identified in the present study. A second limitation of this investigation is that only one buffer was used to react with RuCl_3_, which was PBS. Other buffer systems (e.g., acetate-based systems) should be utilized to further determine if other antivenom compounds can be generated from RuCl_3_ or if physiological pH is key to RuCl_3_-mediated antivenom activity. However, it is possible that RuCl_3_ may interact with many molecules added to buffer to create a pH of 7.4. Thus, it may be difficult to find a physiological buffer that could enhance RuCl_3_ antivenom efficacy without the possibility of interacting with the other molecules present. A third limitation was that only effects on plasma were tested without platelets present. It is entirely possible that *C. atrox* venom could affect platelet function or cause aggregation as has been seen in vivo in rabbits with thrombelastographic methods involving selective platelet inactivation with cytochalasin D [42]. It is possible that *C. atrox* venom may be inactivated by the various RuCl_3_-derived compounds, but perhaps the degree or pattern of platelet inactivation by the venom may be different than that seen with its fibrinogenolytic effects. The best method to address this limitation is to repeat the experimental design with platelets included in whole blood from humans or rabbits using the thrombelastographic methods described [42]. However, such an endeavor is beyond the scope of the present work and should be achieved in the future. Another limitation includes the likelihood of there being several, not just three, compounds formed from RuCl_3_ in the presence of PBS and DMSO, and this study cannot identify them individually. Mass spectrometry would be required to identify the various compounds, but beyond identification, determination of whether each compound or combination of compounds is required to achieve antivenom activity is also necessary. An important limitation also includes the utilization of crude *C. atrox* venom, which exerts its anticoagulant activity via the action of several metalloproteinase and serine protease enzymes. The relative percentage of effect each enzyme has is unknown, and studies that purify and identify individual venom enzymes in sufficient quantity to be investigated are required. Put another way, sorting out the species of RuCl_3_-generated compound, separately or in combination, and matching them to individual enzymes will require enormous resources and time. Such enterprises are far beyond the scope of the present investigation. In summary, this investigation has limitations, but they can be addressed prospectively in future investigations by laboratories equipped with the expertise required.

This study used thrombelastography to document the various biochemical phenomena required to identify the antivenom effects of at least three RuCl_3_-derived compounds. Machinery aside, it is the parameters used (e.g., TMRTG, MRTG, TTG) that made this facile. These parameters are parametric and Newtonian in nature, unlike common clinically used (and used in other laboratories) measures that are not parametric. Traditional variables include the reaction time (R, minutes), which is defined as the time to 2 mm of clot resistance as a measure of the very beginning of coagulation; alpha (or angle, degrees), which is defined as the angle from R to the inflection point of most rapid increase in clot resistance when the rate of clot growth begins to slow; and maximum amplitude (MA, mm), which is the measure of final clot resistance when thrombus growth is completed. This investigator prefers TMRTG to R as TMRTG provides information concerning the greatest velocity of clot growth, which kinetically is of greater interest than simple commencement. Even more critical to assessing changes in coagulation kinetics is the experimentally proven nearly exponential increase in the velocity of clot growth and strength in Newtonian terms when using alpha or MA instead of MRTG and TTG as experimentally demonstrated [43]. Degrees and matching dynes/cm^2^/second measurements were compared with hundreds of observations [43], with the nonlinear relationship modeled. With this data, it was determined that a change in the angle from 40° to 80° resulted in a 16-fold increase in velocity defined as dynes/cm^2^/second. Further, a change in MA from 35 mm to 70 mm resulted in a 4-fold increase in clot strength as defined by TTG in dynes/cm^2^. Thus, measuring and reporting coagulation kinetic data in parametric, Newtonian units is far preferable to more sensitively and parametrically document experimental results.

In conclusion, RuCl_3_ has been demonstrated to be a potent antivenom in vitro [31,32] and a component of a complex, Ru-based antivenom that significantly inhibits or abrogates the effects of hemotoxic venom in a preclinical model of envenomation [33,34]. RuCl_3_ appears to act directly as an antivenom agent and can form additional antivenom agents involving a phosphate- or DMSO-containing compound. Thus, the inorganic/organic antivenom used to markedly inhibit or abrogate the effects of envenomation in a rabbit model [33,34] involves at least four different compounds: (1) a CORM-2-derived ruthenium radical; (2) a RuCl_3_ cation; (3) RuCl_3_-phosphate ion(s); and (4) a RuCl_3_-DMSO compound. The present study provides further insight as to the specific molecular mechanisms at play, with RuCl_3_-generated cations and anions in the absence or presence of phosphate and DMSO presenting varying potencies as antivenom agents. It must be emphasized that this investigation is preliminary—to the degree possible with the methods offered by this laboratory, phosphate buffer and DMSO have been determined to clearly enhance the antivenom activity of RuCl_3_. Additionally, more involved molecular investigations will be required to characterize novel compounds, potential microemulsions, or other, unforeseen findings to explain the critical phenomena identified thus far. Without the findings of the present study, there would be no rationale or interest, from a treatment standpoint, to refine and define the RuCl_3_-derived compounds formed by this unique antivenom. Future investigation is warranted and justified to further define if additional compounds can be generated from RuCl_3_ to act as antivenoms.

## 4. Materials and Methods

### 4.1. Plasma, Chemicals, and Venoms

Pooled normal human plasma that was sodium citrate-anticoagulated and maintained at −80 °C was obtained from George King Bio-Medical (Overland Park, KS, USA). Lyophilized *C. atrox* venom was provided by the National Natural Toxins Research Center (NNTRC) located at Texas A&M University-Kingsville, Kingsville, TX, USA. The National Institutes of Health fund the NNTRC out of the Office of Research Infrastructure Programs. Venom was dissolved into 0.9% NaCl or calcium-free phosphate-buffered saline (Millipore Sigma, Saint Louis, MO, USA) to a final 30 mg/mL concentration, aliquoted, and maintained at −80 °C. RuCl_3_ and DMSO were obtained from Millipore Sigma (Saint Louis, MO, USA). Tissue factor for activating coagulation was obtained in the form of Pacific Hemostasis™ Prothrombin Time Reagent (Thermo Fisher Scientific, Pittsburgh, PA, USA). Calcium chloride (200 mM) was obtained from Haemonetics Inc. (Braintree, MA, USA).

### 4.2. Sample Composition, Incubations, and Coagulation Monitoring

Sample mixtures were placed in a disposable cup in a computer-controlled thrombelastograph^®^ hemostasis system (Model 5000; Haemonetics Inc., Braintree, MA, USA) at 37 °C. The mixture used in the series of experiments was composed of 320 µL of plasma, 10 µL of tissue factor (0.1% final concentration of Pacific Hemostasis™ Prothrombin Time Reagent, Thermo Fisher Scientific, Pittsburgh, PA, USA), 6.4 µL of PBS, 3.6 µL of venom solution, and, lastly, 20 µL of calcium chloride. Plasma, tissue factor, and PBS were placed in the disposable cup, with venom solution added for 1 min prior to the final addition of calcium chloride. Venom solutions were composed of 300 µg/mL venom in NS or PBS that had no addition or had RuCl_3_ placed as a 1% addition for a final concentration of 100 µM RuCl_3_. Experiments involving DMSO had a 1% addition of it to NS (*v*/*v*). After the venom solutions were incubated at room temperature for 5 min, the aforementioned addition to the plasma mixture of the venom mixture was performed and the sample was mixed. Thus, the final concentration of venom in plasma was 3 µg/mL, which was expected to cause fibrinogenolysis as the activity of the enzymes involved are calcium-independent [38]. This brief exposure was designed to result in a reproducible compromise of coagulation that could be quickly assessed by the rapid onset of tissue factor-initiated clot formation. After calcium chloride and final mixing of the samples, the following parameters were determined: time to maximum rate of thrombus generation (TMRTG), which is the time interval (minutes) observed prior to the maximum speed of clot growth; maximum rate of thrombus generation (MRTG), which is the maximum velocity of clot growth observed (dynes/cm^2^/second); and total thrombus generation (TTG, dynes/cm^2^), the final viscoelastic resistance observed after clot formation. Data were collected for 15 min.

### 4.3. Statistical Analyses

Data are presented as means + SD. All conditions were represented by *n* = 6 replicates, as this provided a statistical power ≥ 0.8 with *p* < 0.05 using this methodology to assess differences in thrombelastographic parameters. A commercially available statistical program was used for one-way analyses of variance (ANOVA), followed by Holm–Sidak post hoc analyses (SigmaStat 3.1; Systat Software, Inc., San Jose, CA, USA). Graphics were generated with commercially available programs (Origen 2024, OrigenLab Corporation, Northampton, MA, USA and CorelDRAW 2023, Alludo, Ottawa, ON, Canada). *p* < 0.05 was considered significant.

## Figures and Tables

**Figure 1 ijms-25-06426-f001:**
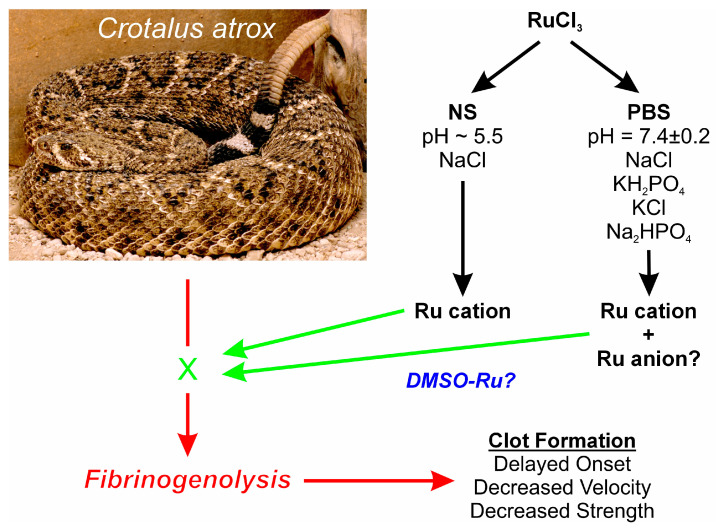
Proposed experimental design. Venoms from the species displayed are placed in NS, PBS, or NS with DMSO, with RuCl_3_ dissolved in water added. Fibrinogenolytic effects of the venom on clot formation (red arrows) detected by thrombelastography could potentially be inhibited by Ru compound ions (green “X” and arrows), and any diluent specific effects on venom activity would also be detected. The photograph of the snake was kindly provided by the National Natural Toxins Research Center at Texas A&M University-Kingsville, Kingsville, TX, USA.

**Figure 2 ijms-25-06426-f002:**
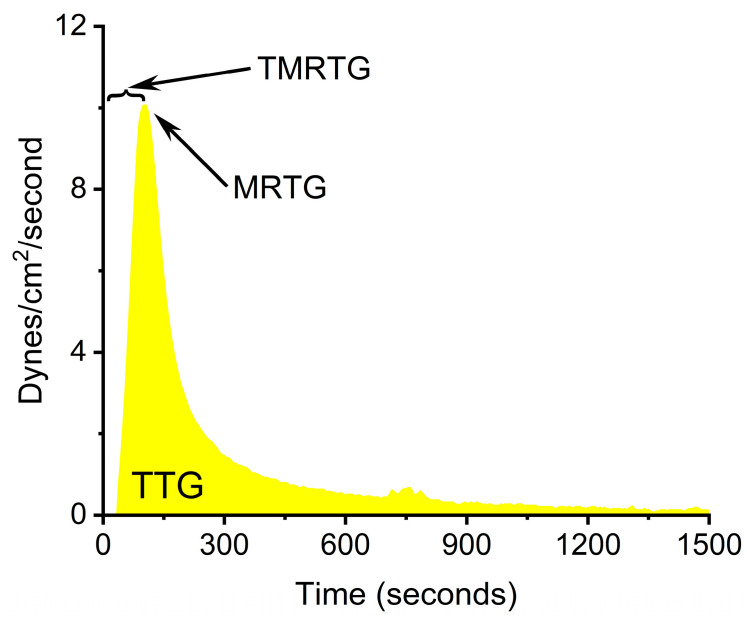
Thrombelastographic parameters displayed as a clot growth velocity curve derived from tissue-activated plasma. TMRTG is the time to the maximum rate of thrombus generation, MRTG is the maximum rate of thrombus generation (dynes/cm^2^/second), and TTG is the total thrombus generation (dynes/cm^2^).

**Figure 3 ijms-25-06426-f003:**
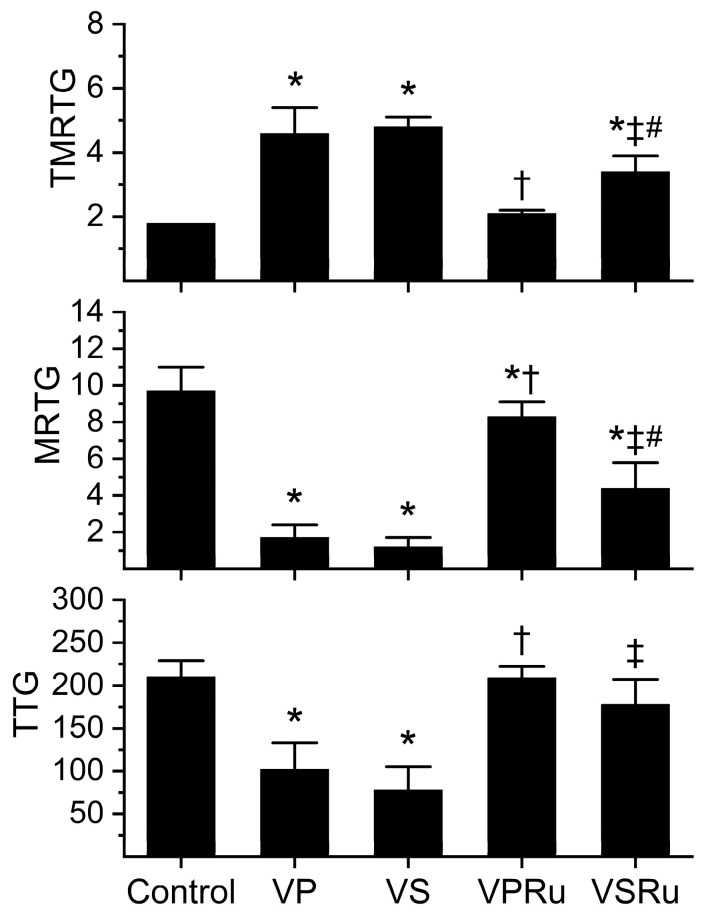
Effects of NS and PBS on RuCl_3_ (100 µM) exposure to *C. atrox* (3 µg/mL)-mediated anticoagulation in human plasma assessed by thrombelastography. Control = plasma without additions; VP = venom in PBS; VS = venom in NS; VPRu = venom and RuCl_3_ in PBS; VSRu = venom and RuCl_3_ in NS (*n* = 6 replicates per condition). Data presented as mean + SD. One-way ANOVA analyses with Holm–Sidak post hoc tests were utilized. * *p* < 0.05 vs. Control; † *p* < 0.05 vs. VP; ‡ *p* < 0.05 vs. VS; # *p* < 0.05 vs. VPRu.

**Figure 4 ijms-25-06426-f004:**
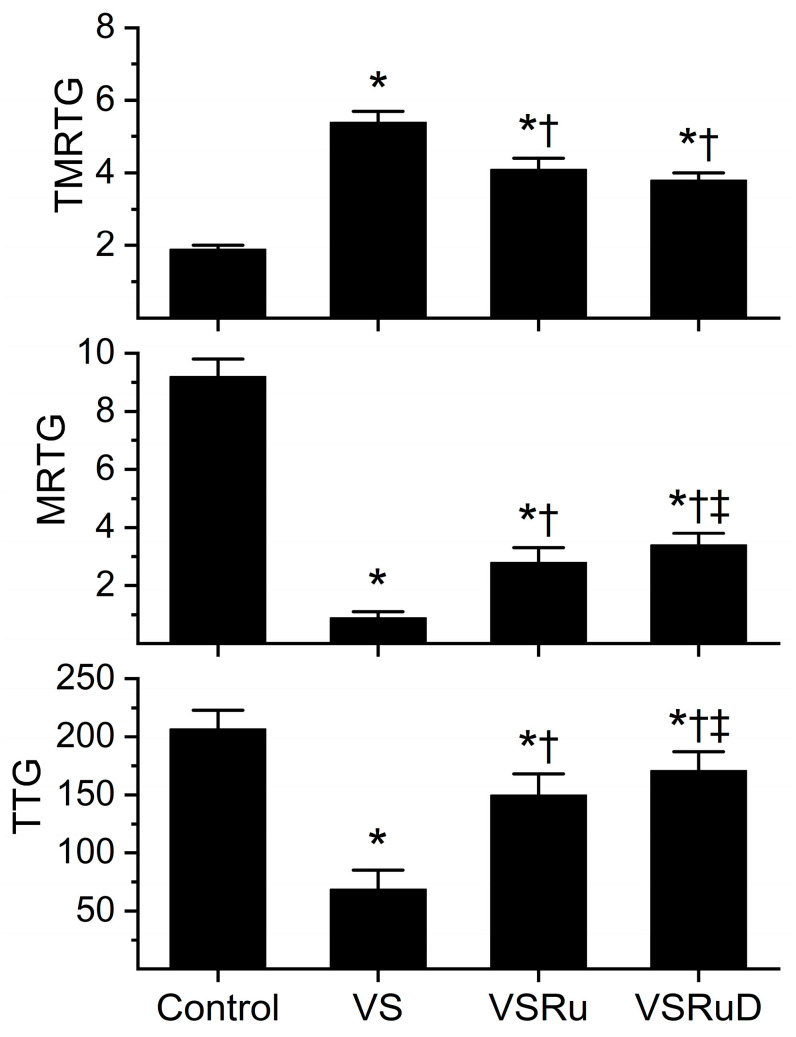
Effect of DMSO in NS on RuCl_3_ (100 µM) exposure to *C. atrox* (3 µg/mL)-mediated anticoagulation. Control = plasma without additions; VS = venom in NS; VSRu = venom and RuCl_3_ in NS; VSRuD = venom and RuCl_3_ in NS with 1% DMSO (*v*/*v*) (*n* = 6 replicates per condition). Data presented as mean + SD. One-way ANOVA analyses with Holm–Sidak post hoc tests were utilized. * *p* < 0.05 vs. Control; † *p* < 0.05 vs. VS; ‡ *p* < 0.05 vs. VSRu.

**Figure 5 ijms-25-06426-f005:**
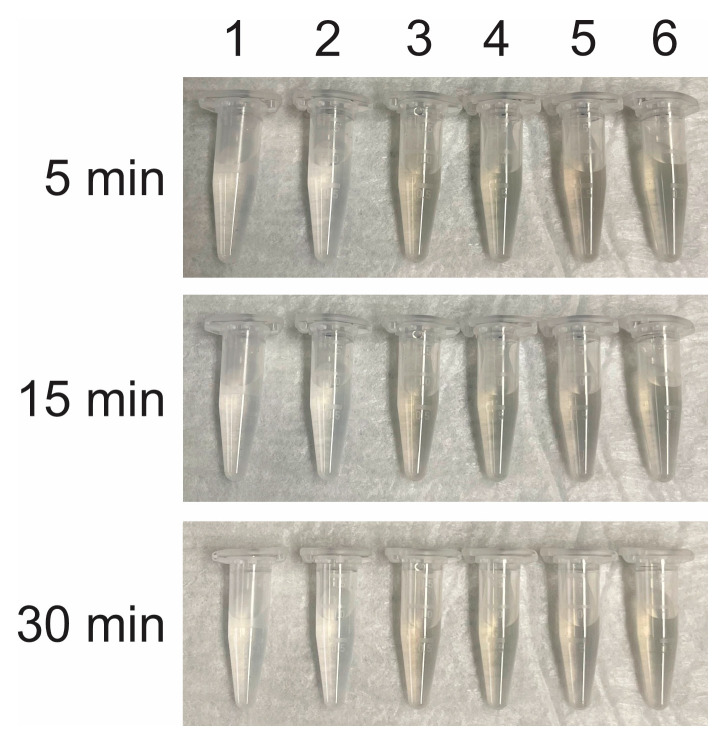
Appearance of NS and PBS with additions at room temperature. 1 = NS + 1% DMSO, 2 = PBS + 1% DMSO, 3 = NS + 100 µM RuCl_3_, 4 = PBS + 100 µM RuCl_3_, 5 = NS + 100 µM RuCl_3_ + 1% DMSO, and 6 = PBS + 100 µM RuCl_3_ + 1% DMSO.

## Data Availability

There is no other data besides that presented in this work.

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
