# Peer review of "Phosphate-Buffered Saline and Dimethyl Sulfoxide Enhance the Antivenom Action of Ruthenium Chloride against Crotalus atrox Venom in Human Plasma—A Preliminary Report"

_ijms, 2024, doi:10.3390/ijms25126426_

Round 1

Reviewer 1 Report

Comments and Suggestions for Authors

Overall an interesting paper, but I feel quite limited in the experimental part. I suggested major revisions because, in my opinion, the authors should address some of the limitations they mention in the discussion. In the current form the manuscript is very speculative in many instances.  Below are also some comments that can help the authors

On page 5, line 161 authors write: "Nevertheless, when considered in composite, these data support the notion that a new, more effective compound formed from RuCl3 and DMSO". This may well be true, but one may also consider the possibility that the 1% DMSO solution actually helps to better solubilize the same resulting Ru complex. It is standard e.g. when evaluating inorganic complexes for their in vitro activities to have stock solutions of the complexes in DMSO and then dilute those to 1% DMSO. In the vast majority of case DMSO just helps with the solubility of the complexes.

In the discussion, page 6, lines 192-200. Authors are principally right in their speculation. This referee has a background in inorganic chemistry, and I would further argue, that not only two, but several different Ru complexes may be formed and at play. RuCl3 may certainly form complexes with- phosphates, sulfur-containing amino acids, histidine or combination of those. Furthermore the pH should have an effect on the nature of the coordinated water molecules (including e.g. an equilibrium of the type [Ru-OH2]n <-> [Ru-OH]n-1 + H+). Thus, yes, “…least two ions derived from RuCl3 in PBS”, but most likely more.

Later (line 201): “Another remarkable finding was that RuCl3 formed a compound containing DMSO”. Possible, but again, please consider the possibility mentioned above. 0.1mM aqueous solutions of RuCl3 may generate micro-suspensions. Thus, although authors admit that it is speculative, caution is advised for this part of the discussion.

Mass spectrometry may help identifying what major species are present in solution by identifying signals with a ruthenium isotope pattern, although I am aware that in such complex mixtures it may be difficult to interpret data. However, authors may consider this possibility.

Minor typos are present (e.g. page 2, line 49); "Another matter to consider is the RuCl3 has..." " ...is that RuCl3 has..."

Comments on the Quality of English Language

Overall the paper reads fine. Minor editing of English language required.

Author Response

“Overall an interesting paper, but I feel quite limited in the experimental part. I suggested major revisions because, in my opinion, the authors should address some of the limitations they mention in the discussion. In the current form the manuscript is very speculative in many instances.  Below are also some comments that can help the authors”

            I appreciate the kind comments of the reviewer. Indeed, I am limited, as I have a predominantly coagulation-heavy laboratory capable of in vitro and in vivo work. However, I do not have the resources to pursue some of the directions the reviewer kindly suggests. I am more than happy to outline the limitations of my work at present and point the way for future work to better define the molecular insights – albeit limited – afforded by what I am capable of achieving.

“On page 5, line 161 authors write: "Nevertheless, when considered in composite, these data support the notion that a new, more effective compound formed from RuCl3 and DMSO". This may well be true, but one may also consider the possibility that the 1% DMSO solution actually helps to better solubilize the same resulting Ru complex. It is standard e.g. when evaluating inorganic complexes for their in vitro activities to have stock solutions of the complexes in DMSO and then dilute those to 1% DMSO. In the vast majority of case DMSO just helps with the solubility of the complexes.”

            The author raises an interesting point, and I was happy to insert this alternative explanation within the paragraph identified. The only issue I have is that RuCl3 is very water soluble. Nothing I have found indicates that this particular, simple Ru-containing molecule needs any additional chemical dissolution to be soluble in aqueous solution, very much unlike CORM-2. As demonstrated in figure 5, nothing seems to be coming out of solution. Nevertheless, I am happy to include the possibility mentioned by the reviewer. I also mentioned the possibility that DMSO in some way enhances the affinity of RuCl3 binding to venom enzymes.

“In the discussion, page 6, lines 192-200. Authors are principally right in their speculation. This referee has a background in inorganic chemistry, and I would further argue, that not only two, but several different Ru complexes may be formed and at play. RuCl3 may certainly form complexes with- phosphates, sulfur-containing amino acids, histidine or combination of those. Furthermore the pH should have an effect on the nature of the coordinated water molecules (including e.g. an equilibrium of the type [Ru-OH2]n <-> [Ru-OH]n-1 + H+). Thus, yes, “…least two ions derived from RuCl3 in PBS”, but most likely more.”

            I appreciate the reviewer’s background and need such insight to explain the possibilities of the various compounds on a molecular level. Thus, I have incorporated some of the text provided by the reviewer in the passage cited. I hope that the reviewer does not mind this, and I really believe it enhances the discussion.

“Later (line 201): “Another remarkable finding was that RuCl3 formed a compound containing DMSO”. Possible, but again, please consider the possibility mentioned above. 0.1mM aqueous solutions of RuCl3 may generate micro-suspensions. Thus, although authors admit that it is speculative, caution is advised for this part of the discussion.”

            I appreciate this cautionary note, and I have modified the indicated passage.

“Mass spectrometry may help identifying what major species are present in solution by identifying signals with a ruthenium isotope pattern, although I am aware that in such complex mixtures it may be difficult to interpret data. However, authors may consider this possibility.”

            I thank the reviewer for this comment and agree. I really do not know where to start, and I do not have the capacity to use mass spectrometry. I also do not know if only one compound or all the compounds in concert are needed for the various effects documented in my system. Such investigation is beyond the scope of my preliminary work, and I ask the reviewer’s indulgence. I address this in the Discussion as a limitation.

“Minor typos are present (e.g. page 2, line 49); "Another matter to consider is the RuCl3 has..." " ...is that RuCl3 has..."”

            Thank you for pointing out this error. I have tried to further refine the text, break up run on sentences and improve clarity.

Reviewer 2 Report

Comments and Suggestions for Authors

Reviewer comments,

The manuscript describes the effect of ruthenium chloride as inorganic antivenom in the presence of phosphate buffered saline or dimethyl sulfoxide to neutralize the coagulopathic action of snake venom in vitro. The author have tested the inhibition of fibrinogenolytic effect induced by Crotalus atrox venom in human plasma. The manuscript is well organized and written. However, I have few comments.

Results

Page 3, the results present two figs 3 on pages 3 and 4. Please, check these.

General comment. The search for inorganic compounds such as RuCl3 with antivenom properties on some viperidae venoms is an interesting topic of research. Thus, as the author mentioned at discussion section, (Pag. 6, lines 184-187), RuCl3 in a cationic form is a significant inhibitor of the fibrinolytic enzymes present in C. atrox venom.

As it is well known, the dominant protein families found in viperidae venoms are, snake venom metalloproteases (SVMPs), snake venom serine proteases (SVSPs) and phospholipases A2 (PLA2s). They appear to be the most toxin proteins in human envenomation, being responsible for coagulopathy, neurotoxicity, myotoxicity and citotoxicity. In connection with this, SVMPs (class P-I and P-III) represent above of 34% of total proteins across all viper species. Moreover, they are fibrino(geno)lytic enzymes involved in the hemotoxic disturbances triggered by these venoms. In regard to this point, it could be interesting to assess the effect of RuCl3 using an isolated SVMP of the class P-I or P-III, and compare the results with those obtained using crude venom. In addition, I suggest to test   the action of RuCl3 on whole blood. It is possible to answer the point raised on page 7 (lines 240-241) to explain the mechanism of action of the inorganic antivenom.  It is relevant to explore the utility of using small molecules as toxin inhibitors, particularly against SVMPs and PLA2s.

Reviewer comments,

The manuscript describes the effect of ruthenium chloride as inorganic antivenom in the presence of phosphate buffered saline or dimethyl sulfoxide to neutralize the coagulopathic action of snake venom in vitro. The author have tested the inhibition of fibrinogenolytic effect induced by Crotalus atrox venom in human plasma. The manuscript is well organized and written. However, I have few comments.

Results

Page 3, the results present two figs 3 on pages 3 and 4. Please, check these.

General comment. The search for inorganic compounds such as RuCl3 with antivenom properties on some viperidae venoms is an interesting topic of research. Thus, as the author mentioned at discussion section, (Pag. 6, lines 184-187), RuCl3 in a cationic form is a significant inhibitor of the fibrinolytic enzymes present in C. atrox venom.

As it is well known, the dominant protein families found in viperidae venoms are, snake venom metalloproteases (SVMPs), snake venom serine proteases (SVSPs) and phospholipases A2 (PLA2s). They appear to be the most toxin proteins in human envenomation, being responsible for coagulopathy, neurotoxicity, myotoxicity and citotoxicity. In connection with this, SVMPs (class P-I and P-III) represent above of 34% of total proteins across all viper species. Moreover, they are fibrino(geno)lytic enzymes involved in the hemotoxic disturbances triggered by these venoms. In regard to this point, it could be interesting to assess the effect of RuCl3 using an isolated SVMP of the class P-I or P-III, and compare the results with those obtained using crude venom. In addition, I suggest to test   the action of RuCl3 on whole blood. It is possible to answer the point raised on page 7 (lines 240-241) to explain the mechanism of action of the inorganic antivenom.  It is relevant to explore the utility of using small molecules as toxin inhibitors, particularly against SVMPs and PLA2s.

Author Response

“The manuscript describes the effect of ruthenium chloride as inorganic antivenom in the presence of phosphate buffered saline or dimethyl sulfoxide to neutralize the coagulopathic action of snake venom in vitro. The author have tested the inhibition of fibrinogenolytic effect induced by Crotalus atrox venom in human plasma. The manuscript is well organized and written. However, I have few comments.”

            I appreciate the reviewer’s kind words, and I hope that my efforts to address the issues raised are acceptable.

Results

“Page 3, the results present two figs 3 on pages 3 and 4. Please, check these.”

            Thank you very much for finding this error. I have corrected it.

“General comment. The search for inorganic compounds such as RuCl3 with antivenom properties on some viperidae venoms is an interesting topic of research. Thus, as the author mentioned at discussion section, (Pag. 6, lines 184-187), RuCl3 in a cationic form is a significant inhibitor of the fibrinolytic enzymes present in C. atrox venom.”

            I appreciate this comment.

“As it is well known, the dominant protein families found in viperidae venoms are, snake venom metalloproteases (SVMPs), snake venom serine proteases (SVSPs) and phospholipases A2 (PLA2s). They appear to be the most toxin proteins in human envenomation, being responsible for coagulopathy, neurotoxicity, myotoxicity and citotoxicity. In connection with this, SVMPs (class P-I and P-III) represent above of 34% of total proteins across all viper species. Moreover, they are fibrino(geno)lytic enzymes involved in the hemotoxic disturbances triggered by these venoms. In regard to this point, it could be interesting to assess the effect of RuCl3 using an isolated SVMP of the class P-I or P-III, and compare the results with those obtained using crude venom.”

            I whole heartedly agree that the reviewer’s proposed studies will be of interest in the future. I do not have the capacity to isolate, purify, and identify with mass spectrometry the individual enzymes contained in Crotalus atrox venom, and I certainly do not have the resources to create enough quantity of the many enzymes to test in my in vitro, plasma-based system. The goal was to use a crude enzyme already known to be inhibited in vitro and in vivo with ruthenium containing antivenoms as a tool to assess variations of RuCl3 compound-generating conditions to search for new antivenoms. That goal was successfully achieved. I have included these suggestions in my Discussion as forward looking investigations.

“In addition, I suggest to test   the action of RuCl3 on whole blood.”

            I have tested the effects of RuCl3 on human plasma as cited in this manuscript (Reference 30: J Thromb Thrombolysis 2021, 51, 577-583.). While interesting, I have no human or animal ethical approvals or protocols for such work. Further, I would have to obtain equipment and expertise to assess effects on platelets in isolation and then factor in the effects of red blood cells. I have already demonstrated that the administration of RuCl3 as a site directed antivenom does not adversely affect whole blood coagulation as demonstrated in the in vivo studies contained within my reference list. Lastly, the systemic administration of RuCl3 as a therapy would never be considered, as doses large enough to have an effect would be toxic. Thus, I ask the reviewer’s indulgence and will not begin such an enterprise as it is outside the scope of the present, preliminary work.

“It is possible to answer the point raised on page 7 (lines 240-241) to explain the mechanism of action of the inorganic antivenom.  It is relevant to explore the utility of using small molecules as toxin inhibitors, particularly against SVMPs and PLA2s.”

            Again, I appreciate and agree with the solution offered by the reviewer. However, the author does not have the capacity to isolate individual venom enzymes, assess the kinetics of fibrinogenolysis of each enzyme in plasma and whole blood, and assess the impact of every potential RuCl3 derived compounds identified by mass spectrometry as an antivenom. Such an enterprise would involve enormous expenditures and years of time to achieve. This is far beyond the scope of the present, preliminary work. I have mentioned such activities as future investigations in the Discussion, and I ask the reviewer’s indulgence in this matter.

It appeared that the reviewer or editorial staff accidently uploaded the comments twice. Therefore, I only addressed them once as previously presented.

Round 2

Reviewer 1 Report

Comments and Suggestions for Authors

Authors have overall considered my comments but only marginally improved the manuscript. I understand and appreciate the comment by the authors regarding their current capabilities, but I am obliged to suggested rejection of the manuscript. As mentioned in my first report, the manuscript is speculative in many instances. In my opinion, to be considered for publication the authors must address some of the limitations they mention in the discussion.

Comments on the Quality of English Language

As indicated above, only minor editing of English language may be required

Author Response

“Authors have overall considered my comments but only marginally improved the manuscript. I understand and appreciate the comment by the authors regarding their current capabilities, but I am obliged to suggested rejection of the manuscript. As mentioned in my first report, the manuscript is speculative in many instances. In my opinion, to be considered for publication the authors must address some of the limitations they mention in the discussion.”

I appreciate the author’s serious comments, and to the best of my ability I tried to address them. I appeal to the reviewer to note that I incorporated the generously provided suggestions, with the equation justifying more possible compounds that I considered of particular interest. I greatly appreciate the background as an inorganic chemist mentioned by the reviewer and offered as much as I could in limitations.

When I wrote this paper, I proposed a limited scope of data outcome in my capacity as an expert in coagulation, and to lesser extent, a toxinologist. I was surprised and delighted that I found what I could with my methods. I was proud to point the ways suggested by the reviewer for future work and did not argue with any of it. Nevertheless, without that first report of the phenomena of interest, there cannot be further, more sophisticated endeavors to more definitively define all molecules involved.

I have modified the title of the article to reflect this:

“Phosphate Buffered Saline and Dimethyl Sulfoxide Enhance the Antivenom Action of Ruthenium Chloride Against Crotalus atrox Venom in Human Plasma – A Preliminary Report”

I have also modified the last paragraph of the manuscript to again emphasize the preliminary nature of my work:

“…It must be emphasized that this investigation is preliminary – to the degree possible with the methods offered by of this laboratory, phosphate buffer and DMSO have been deter-mined to clearly enhance antivenom activity of RuCl3. Additional, more involved molecular investigations will be required to characterize novel compounds, potential microemulsions, or other, unforeseen findings to explain the critical phenomena identified thus far. Without the findings of the present study, there would be no rationale or interest, from a treatment standpoint, to refine and define the RuCl3 derived compounds formed by this unique antivenom. Future investigation is warranted and justified to further define if additional compounds can be generated from RuCl3 to act as antivenoms.”

In concluding, I implore the reviewer to reconsider the decision. I have made a best faith effort. More investigation can always be performed by others, and should be. I simply cannot go beyond my capacity – and I have couched my conclusions with limitations the reviewer suggested without hesitation. Thank you for considering my work.

Round 3

Reviewer 1 Report

Comments and Suggestions for Authors

Daer Author. I appreciate your effort and reply. Just to be clear, I have nothing against publication of the study, and indeed, as you may see, I have recomended now "accept after minor revision". As previoulsy mentioned, I found the study interesting overall, but I always felt that my role as a reviewer is to make sure that all angles of a study are considered and advice consequently the editorial office if the science is sound with no "stones unturned". In that sense I meant and wrote that the study was speculative in some parts. Indicating "A Preliminary Report" in the title I think it is fair, as it is the added paragraph in the discussion. Both helped me to reconsider my recomendation. The study as such can be now considered as phenomenological (not in the philosophical sense), which is fine. I leave now to the editors the final decision. My best regards.

Comments on the Quality of English Language

As previously.

Author Response

“Daer Author. I appreciate your effort and reply. Just to be clear, I have nothing against publication of the study, and indeed, as you may see, I have recomended now "accept after minor revision". As previoulsy mentioned, I found the study interesting overall, but I always felt that my role as a reviewer is to make sure that all angles of a study are considered and advice consequently the editorial office if the science is sound with no "stones unturned". In that sense I meant and wrote that the study was speculative in some parts. Indicating "A Preliminary Report" in the title I think it is fair, as it is the added paragraph in the discussion. Both helped me to reconsider my recomendation. The study as such can be now considered as phenomenological (not in the philosophical sense), which is fine. I leave now to the editors the final decision. My best regards.”

I greatly appreciate the comments of the reviewer. I have looked carefully at these final comments, and it appears that the changes I have made are satisfactory. I do not see anything else I can do with the text, so I am submitting my manuscript once again for final consideration by the academic editor. I again thank the reviewer for assisting me with improving my work to the extent possible with the data presented. With all sincerity.